# Involuntary temporary work and mental health medications: A longitudinal study in Denmark

**Karsten Albæk** *, **Stefan Bastholm Andrade** *

VIVE–The Danish Center for Social Science Research, Copenhagen K., Denmark

* kal@vive.dk (KA); sba@vive.dk (SBA)

## Abstract

Since the 1970s, most western countries have experienced an increase in jobs characterized by temporary employment working arrangements. Research links temporary employment to negative health outcomes. Yet, no study has analysed the effects on the mental health of workers in involuntary temporary employment. This study analyses the consequences of involuntary temporary employment for mental health. We distinguish between different lengths of exposure to involuntary temporary employment and assess the effects separately for women and men. We use a cohort design that combines data from the Danish version of the longitudinal European Labour Force Survey with administrative data about use of prescription drugs for anxiety and stress. Using a fixed effects approach, we identify the effects of involuntary full-time temporary employment on mental health over time. To further investigate causal effects, we also compare the outcomes of workers in involuntary full-time temporary employment with a control group that consists of workers who become employed in involuntarily full-time temporary job at a later point of time. For women in involuntary full-time temporary employment (for six quarters or more) the results show a deterioration in mental health as indicated by a 12.8 percentage point increase in drug use. Involuntary full-time temporary employment for one quarter results in a decrease in drug use by 1.1 percentage points, but no lasting effects. For men, we find no mental health consequences of involuntary full-time temporary employment. We conclude that involuntary full-time temporary employment for six quarters or more is likely to be harmful for women's mental health, while shorter periods of involuntary full-time temporary employment may have a minor positive impact. The implications suggest that it is advisable for labour market policy to make it easy for those in involuntary temporary employment to find permanent jobs, and that policy makers should consider adverse mental health problems when addressing policies affecting the prevalence of temporary employment.

## Introduction

Compared to workers in permanent employment, workers in temporary employment tend to have lower and more unstable incomes [1]. In addition to the economic distress, research links

**Data Availability Statement:** Due to restrictions related to Danish law and protecting citizen privacy, the combined set of data as used in this study can only be made available through a trusted third party, Statistics Denmark. University-based Danish

scientific organizations can be authorized to work with data within Statistics Denmark and such organization can provide access to individual scientists inside and outside of Denmark. Requests for data may be sent to Statistics Denmark or the Danish Data Protection Agency. The LFS data are open access and permission to use these data may be granted by Eurostat. The procedure and conditions for approval is described here: https://ec.europa.eu/eurostat/documents/203647/771732/How_to_apply_for_microdata_access.pdf.

**Funding:** This work was supported by the Danish Working Environment Research Fund (grant no. 15-2019-09 to KA and SBA). The funder had no role in study design, data collection and analysis, decision to publish, or preparation of the manuscript.

**Competing interests:** The authors have declared that no competing interests exist.

temporary employment to poor working environments, lack of training, bleak labour market prospects, and poor health [2, 3]. A particular concern is whether job insecurity associated with this type of employment may also lead to psychological distress [4, 5]. Drawing on cross-sectional data, studies show that, on average, workers in temporary employment have poorer mental health than workers in permanent employment [6, 7].

However, the relationship between temporary employment and mental health may be bidirectional: While temporary employment can lead to mental health problems, workers with mental health problems might be more likely to enter temporary employment. To address the issue of whether temporary employment causes poor mental health, recent studies draw on longitudinal data. The findings of these studies are inconclusive. Two studies using Australian longitudinal data find no connection between temporary employment and mental health [8, 9], and one study using UK data finds that temporary work arrangements have no long-lasting detrimental health consequences [10]. For Italy, a study finds that temporary employment has adverse consequences for mental health [11], and the same is the case for a study on the US [12].

In this study, we make four contributions to the literature on temporary employment and mental health. First, our study is the first to conduct analyses of the consequences of *involuntary* temporary employment (ITE) on mental health. We use data from the Danish version of the longitudinal European Labour Force Survey (LFS), which we merge with administrative data on drugs prescribed for anxiety and stress. We focus on those survey respondents who answered that they have employment contract of limited duration because they could not find a permanent job, i.e., the involuntary temporary employees. Second, our study is the first to examine the extent to which the length of exposure to temporary employment plays a role for mental health. Third, we report gender specific results, which are rarely found in the literature on temporary employment and health [13]. Fourth, we draw on recent innovations in causal inferences to estimate the causal effect of temporary employment on mental health problems [14–16].

## Background

### Temporary employment in a Scandinavian context

Since the 1970s, western countries have experienced an increase in jobs with "non-standard", "atypical", "precarious" or "flexible" working arrangements [17, 18]. For example, data for the European countries show that while 8 pct. of the labour force was in temporary employment in 1983, the share of workers had risen by 75 pct. to 14 pct. of total European employment in 2017 [17]. Research explains the rise in non-standard working arrangements with references to globalization, a decrease in the role of labour unions, technological changes, "fissuring" of traditional workplaces as work processes are outsourced, and labour market reforms involving changes in employment protection [1, 19–21].

Studies suggest that the distressing effects of temporary work might vary across welfare states [3, 22]. For example, there is some evidence that Scandinavian-type welfare systems provide protection against adverse effects of flexible employment relationships [23]. Compared to other western labour markets, the Danish labour market is characterized by substantial *flexibility*, with high levels of job and worker flows that are comparable to the U.S. level [24]. This labour market arrangement implies that displaced workers find new jobs with comparative ease [25]. Furthermore, *security* in Denmark is high because the social democratic welfare state keeps income maintenance comparatively high [26]. This combination concept, *flexicurity*, has played a large role in the development of European policies [27] and in the academic debate about precarious work [28, 29].

### Literature on temporary employment and mental health

The relationship between temporary jobs and health is bidirectional: temporary employment might lead to health problems but workers with health problems might be more likely to enter temporary employment. A substantial share of the early contributions to the literature assess the health of temporary workers relative to workers with permanent contracts using cross-sectional data. A recent review lists 39 cross-section studies and three longitudinal studies on the association between fixed-term employment and mental health ([6] see also [7]). In the following, we review recent contributions on the relationship between temporary employment and health, where the health status of workers is followed over time.

Bardasi and Francesconi [10] apply ten waves of the British Household Panel Survey (BHPS) to study the relationship between different types of employment contracts and mental health. The descriptive statistics show no statistically significant differences in mental health between workers on temporary and permanent contracts for women and men. When applying fixed effects estimation, the authors find that temporary work arrangements did not have long-lasting detrimental health effects on either female or male workers. However, as the authors note, they cannot distinguish between involuntary and voluntary temporary employment and they state that health effects may even be positive in the latter case.

Quesnel-Vallée et al. [12] analyse the effect of temporary work using answers from six biannual versions of the US National Longitudinal Survey of Youth 1979. They compare the outcome for workers in temporary employment with a control group constructed by propensity score matching. The authors find a substantial and significant increase in the severity of depression symptoms associated with exposure to temporary work at any point in time in the two preceding years.

Dawson et al. [30] use 18 waves of the annual British Household Panel Survey (BHPS) to analyse the extent to which workers who enter temporary employment differ from other groups with respect to health, where mental health is assessed by answers to questions about psychological distress, psychological anxiety and life dissatisfaction. The authors find that permanent contract employees who transit into temporary employment have lower levels of mental health relative to individuals who never transition into temporary employment.

Pirani and Salvini [31] use four years of the Italian version of the European Union Statistics on Income and Living Conditions survey. Their outcome variable is answers to the question "How is your health in general?" The respondents are considered to have health problems if the answers are fair, poor or very poor. In addition to regressions on the survey data, the authors apply inverse probability weights calculated from propensity score matching to adjust for the probability of entering temporary employment. The authors find that temporary employment does not entail significant adverse consequences for men, but is strongly harmful for Italian women. Temporary employment becomes particularly negative for health when it is prolonged over time, while a temporary contract followed by a permanent one within "a reasonable lapse of time", e.g, one year, has no negative consequences for the worker's health.

Moscone et al. [11] use information on workforce contract type for five years of administrative data for the Lombardy region of Italy. Information on mental health stems from administrative registers on prescription drugs, and the authors construct a variable that takes the value one if a person received at least one medical prescription for two or more consecutive quarters (i.e., 6 months). The empirical model includes lagged health status and control functions, where firm level variables for contract type are included as instruments. The authors find that being in a state of temporary employment tends to increase the likehood of developing mental health problems; moving from a permanent to a temporary contract tends to deteriorate mental health, while a shift from a temporary to a permanent contract improves mental health status.

Bender and Theodossiou [32] apply 17 waves of the BHPS, where the employed respondents are categorized as having either permanent or flexible employment contracts. The outcome variables are subjective evaluations of whether health was excellent or good, and answers to questions about specific health problems, such as anxiety, depression, heart or blood pressure problems, stomach or digestion problems. In the analysis, the authors only include those individuals who state that they are in good health and in permanent employment in the initial period. They use hazard regressions, including a frailty distribution, to take account of unobserved heterogeneity and conclude that individuals with long spells of "atypical" employment have higher levels of stress.

In summary, the reviewed studies performed using longitudinal data yield the impression that workers on temporary contracts experienced health problems before they were in temporary employment. However, some studies also presented evidence that temporary employment worsens health problems, at least for longer periods of temporary employment. The single study with separate analyses for women and men finds adverse consequences for women but not for men.

## Methods

### Data

The data come from the European Union Labour Force Survey (LFS), which is a quarterly comparative panel survey that interviews a sample of people in the European countries about their labour market participation. The respondents are interviewed up to four times. The second interview takes place one quarter after the first survey, while the third and fourth interviews take place after a two-quarter pause. In our analysis, we use interview rounds collected between 2006 and 2018 and limit our sample to respondents between ages 20 and 64.

Fig 1 illustrates the data collection in LFS for the first quarter of 2006 to the last quarter of 2008. The first group, labelled group A, is interviewed in the first and the second quarter of 2006. After a break during the third and fourth quarter of 2006, the group is interviewed again in the first and the second quarter of 2007.

Group B starts the first interview in the second quarter of 2006 and is interviewed for the last time in the third quarter of 2007. The LFS continues to add new respondents in each new quarter, not only for the groups C to H that are included in Fig 1 but up to 2018, which is the last year of our data.

In our main analysis, we include all respondents who have answered at least once that they are in ITE. If a respondent states they are in ITE in the first quarter of the LFS, we do not have information about their previous labour market status. We complement the main analysis with an analysis of how the consequences of ITE might differ depending on the worker's previous labour market status. This complementary analysis demands that a labour market state is observed before the first quarter in which respondent answers that she is in ITE. As a result, the complementary analysis only includes respondents who have participated a maximum of three times in the LFS.

### Involuntary temporary employment

Since 2006, participants in the LFS have been asked whether they are employed on a temporary contract and, if so, the reason for their temporary employment. The participants with temporary employment can state one of the following four reasons: they could not find a permanent job, they did not want a permanent job, they have a training contract, or they have a contract for a probationary period. Our treatment group consists of full-time employees who answered that they were on a temporary contract because they could not find a permanent job. We refer

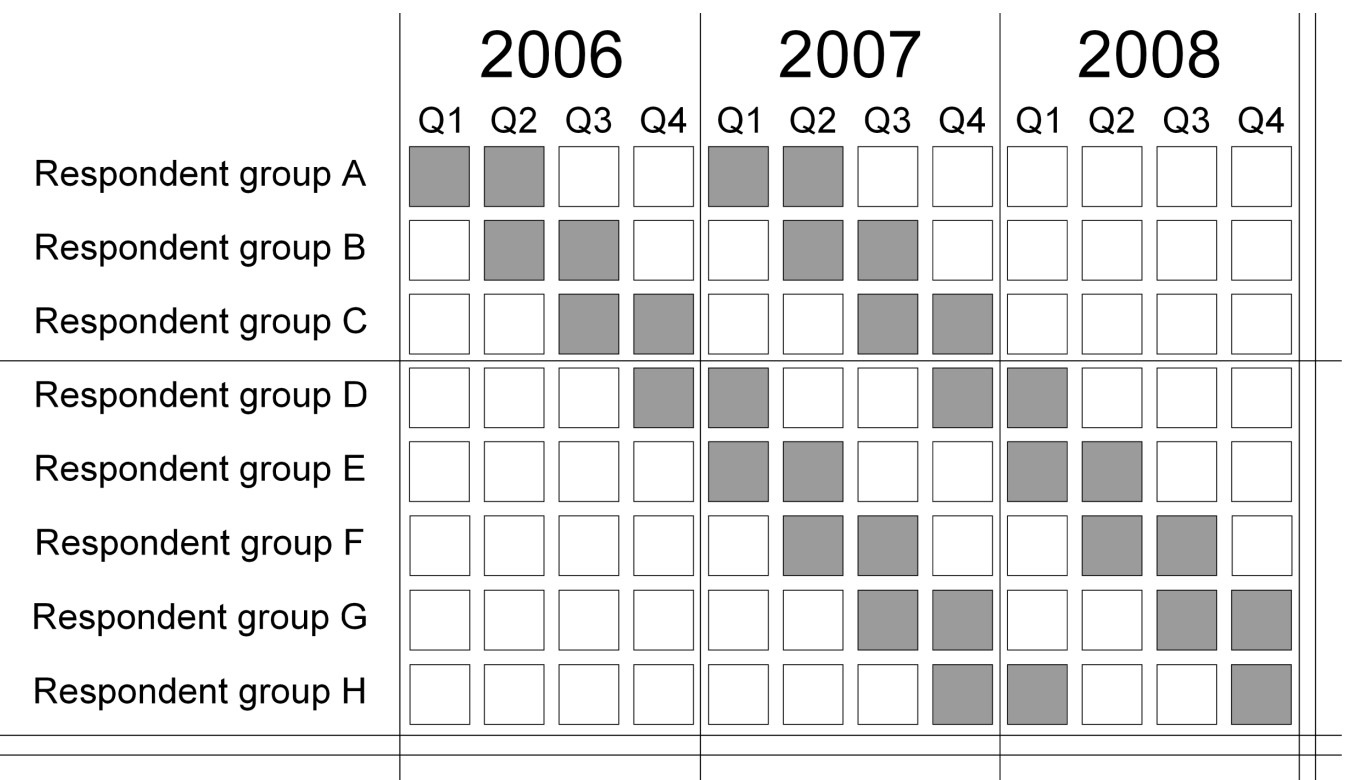

**Fig 1. Illustration of the data collection process for the Labour Force Survey for eight rounds of interviews between 2006 and 2008, where grey boxes indicate that the respondent group is invited for an interview.**

to this group as "involuntary temporary workers". In addition, we construct a comparison group consisting of all other respondents who stated that they were employed full-time on a permanent contract. We label this group the "permanent contract workers." In our analyses, we include variables that measure different lengths of ITE varying from one to six quarters of employment.

We confine the analysis to workers in full-time involuntary temporary employment only. Our reason for not including part-time temporary workers is that full-time workers might exhibit higher levels of organizational commitment and job satisfaction compared to part-time employees. As a result, mental health consequences of temporary employment might also differ for full-time and part-time workers. Inclusion of part-time workers in the analysis would thus make our results a mixture between the responses of the two groups of workers.

### Medication for certain mental health problems

We supplemented the LFS with detailed administrative data on the respondents' health status from the Prescription Drug Database that covers all prescribed drugs purchased by Danish residents. The database contains a detailed classification of medicine according to the Anatomical Therapeutic Chemical (ATC) system. We construct a binary indicator for mental health that measures whether the respondent has been prescribed one or more drugs for mental health problems during a quarter. We follow the mental health of the participants for eight quarters before the first survey and eight quarters after the last survey. We use the same two groups of drugs as indicators for mental health as in Rocco et al. [33]: anti-hypertensive drugs and psychotropic drugs. Anti-hypertensive drugs include ATC-codes C02, C03, C07, C08, and C09,

while the psychotropic drugs include ATC-codes N05 and N06. While a number of studies have used medication as a measure of mental health problems, we are aware that many people with similar health problems (for various reasons) do not use medication. Our analyses should, therefore, be seen as relatively conservative estimates, as the number of people with mental health problems due to ITE may be significantly higher.

## Control variables

Inspired by previous research [29], our regression models include control variables for the workers' education, occupation and income. We draw on the International Standard Classification of Education to classify the workers' education into three categories: "high" (university or college), "middle" (vocational or high school), and "low" (below high school). The occupational categories are white-collar workers (main categories 0–4 in the International Standard Classification of Occupations, ISCO), and blue-collar workers (main ISCO categories 5–9). Information about income comes from tax registers. From these registers, we apply a variable measuring the annual total income that includes earnings and capital income. We also include demographic household variables about family composition (i.e., marital status, an indicator for children below age 6 living in the household, and an indicator for children below age 18 living in the household).

## Ethics statement

The participants in the LFS provide consent to the administrators of the survey. The LFS data was merged with administrative data at Statistics Denmark. Researchers at approved research institutions have access to these data for research purposes under confidentiality rules.

## Methodology and empirical strategy

We use variants of regression analysis to determine the effect of the treatment variable (ITE) on our outcome variable (medication for certain mental health problems). The association between treatment and outcome is measured by coefficients on indicator variables taking the value one if the respondent is in temporary employment in a quarter, and zero otherwise. We code different lengths such that there is one indicator for being in involuntary temporary employment for one quarter and other indicators for being in temporary employment for two, four, five and six quarters (the pause in collecting the survey implies that spells of three quarters of temporary employment are not observed in the data). In addition, we apply indicator variables before and after treatment to assess pre- and post-treatment levels of mental health.

We estimate three types of models. First, we compare the outcome of the treatment group with the comparison group of other full-time employees using a linear regression model (OLS). Second, we use the treatment group to estimate regression models containing separate intercepts for each participant, i.e., fixed effects. Fixed effects models exclude effects of time-invariant factors that might influence the workers' mental health. We therefore only include time varying explanatory variables in these models. Third, we apply "difference-in-difference matching" [16] by forming a control group that consists of workers who enter temporary employment in a later quarter [14, 15]. This procedure has the potential to alleviate the impact of unobserved differences between workers on the outcome variable.

We use sample weights for calculating all the statistics that are reported in the paper. The sample weights of the LFS take both stratification and attrition into account. For each respondent, we apply the average of the sample weights in the surveys. We handle missing values as follows: Respondents without educational information are omitted from the analyses, corresponding to 2.3 pct. of the observations; we assign workers zero children if there is no

information about children in the data; workers without marriage information are treated as singles; six workers without occupational information are categorized as blue-collar workers on the basis of their educational information.

While our main focus is on involuntary full-time temporary employment, we also include workers with part-time employment in the subsection of the analysis entitled "Involuntary temporary employment and previous labour force states". In this part of the analysis, we assess the role of previous labour market states before the worker enters involuntary full-time temporary employment, which might include part-time employment.

## Results

### Descriptive statistics

Table 1 presents summary statistics (mean and standard deviation) for the variables in the analysis, separately for women and men. The treatment groups of workers in full-time temporary employment consist of 5,365 women and 3,827 men. Among the group of women, 14 pct. received a prescription for mental health drugs in contrast to 17 pct. of the women in the comparison group with permanent full-time employment. For men, the corresponding figures are lower at 12 pct. and 13 pct. For both women and men, the average age of temporary workers is lower than workers in the comparison group (37–38 years compared to 42 years). The lower age of temporary workers is reflected in a lower marriage rate and lower shares of workers with children. The magnitude of the standard deviation of age shows that temporary employment is prevalent among different age groups of the workforce (and not confined to, e.g., younger age groups). Older full-time workers are more frequent users of prescription drugs

**Table 1. Mean and standard deviation of variables, full-time workers of age 20–64 in involuntary full-time temporary employment (treatment group) and permanent full-time employment (comparison group).**

| | Women | | | | Men | | | |
|---|---|---|---|---|---|---|---|---|
| | Involuntary temporary employment | | Permanent employment | | Involuntary temporary employment | | Permanent employment | |
| | Mean | Std.dev | Mean | Std.dev | Mean | Std.dev | Mean | Std.dev |
| Prescription drugs for mental health problems | 0.142 | 0.355 | 0.165 | 0.371 | 0.118 | 0.333 | 0.128 | 0.333 |
| Education level | | | | | | | | |
| Low | 0.146 | 0.359 | 0.152 | 0.358 | 0.214 | 0.423 | 0.198 | 0.398 |
| Middle | 0.391 | 0.495 | 0.471 | 0.499 | 0.464 | 0.516 | 0.552 | 0.497 |
| High | 0.464 | 0.506 | 0.377 | 0.484 | 0.323 | 0.484 | 0.251 | 0.433 |
| White collar | 0.526 | 0.507 | 0.540 | 0.498 | 0.440 | 0.512 | 0.475 | 0.499 |
| Married | 0.387 | 0.494 | 0.549 | 0.497 | 0.327 | 0.484 | 0.531 | 0.498 |
| Children up to 6 years old | 0.018 | 0.134 | 0.020 | 0.140 | 0.006 | 0.080 | 0.020 | 0.140 |
| Children up to 18 years old | 0.216 | 0.418 | 0.290 | 0.453 | 0.144 | 0.362 | 0.250 | 0.432 |
| Income, in 1000s DKK, 2018 level | 317 | 138 | 387 | 159 | 333 | 242 | 460 | 373 |
| Age | 36.8 | 11.4 | 42.1 | 11.0 | 37.9 | 12.5 | 41.9 | 11.4 |
| Year | 2012.0 | 3.7 | 2011.8 | 3.9 | 2012.1 | 3.6 | 2011.7 | 3.9 |
| No. of persons | 5,365 | | 67,142 | | 3,827 | | 80,216 | |

*Notes*: Based on persons who had full time employment in at least one of the LFS surveys, 2006–2018. The treatment group, "Temporary", had temporary employment in at least one of the surveys. The comparison group, "Permanent," had a permanent contract in at least one of the surveys (and no temporary employment in the others). Levels of variables are assessed at the quarter of the first observation of temporary employment and the quarter of the first observation of a permanent contract, except health for the treatment group, which is assessed two quarters before the first observation of temporary employment. Rows 2 to 8 display statistics for indicator variables that take the value one if the respondent belongs to the stated group and zero otherwise. The income variable is the yearly personal income deflated to 2018 Danish kroners (one Euro is about 7.4 Danish kroners). Sample weights are used.

than younger ones: The share of women using prescription drugs increases from 6 pct. among workers aged 20–24 to 38 pct. among those aged 60–64, while the corresponding shares for men are 3 pct. and 36 pct.

## Mental health before, during, and after involuntary temporary employment

Fig 2 illustrates the pattern of drug use for mental health problems before, during, and after ITE separately for women (Panel A) and men (Panel B). For women, drug use during the first four quarters of temporary employment (Graph A2) tends to be lower than drug use in most of the period before entering temporary employment (Graph A1), while workers on spells of five and six quarters of temporary employment use more drugs than before entering temporary employment. After temporary employment (Graph A3), drug use returns to the pre-treatment level. For men on temporary employment (Graph B2), drug use during most of the quarters appears to be higher than before entering temporary employment (Graph B1).

While all workers with ITE are included in the coefficient on the first treatment indicator (one quarter), only workers with six quarters of ITE enter the calculation of the last treatment indicator. As a result, the confidence intervals in Fig 2, Graphs A2 and B2 increase as temporary employment periods become longer. The height of the dots in the diagrams indicates the level of mental health in the treatment group relative to the comparison group, taking age differences into account. The levels in Fig 2, Graphs A1 and B1 show that before entering temporary employment workers, on average, used more drugs on average than workers with permanent employment.

## Regression results

Table 2 shows the results of regression models for women (columns 1 and 2) and men (columns 3 and 4). The model in the column 1 is based on an OLS estimation on data for women from both the treatment group and the comparison group of permanent contract workers. The coefficient on the indicator for belonging to the treatment group is positive and significantly different from zero at the 1 pct. level, which means that women in ITE use prescription drugs more frequently than those with permanent employment.

Furthermore, the model in column 1 shows that the first quarter of ITE for women is associated with a decline in drug use and the same is the case for the last quarter before entering temporary employment. ITE of four quarters of temporary employment also appears to be associated with improved mental health for women. Furthermore, ITE does not appear to have lasting effects on prescription drug use for mental health (the coefficient on the indicator of post-treatment drug use is close to zero and not significantly differently from zero).

Model 2 in column 2 only includes women with ITE and uses fixed effects estimation on longitudinal data. In contrast to Model 1 in column 1, we no longer compare the effects with workers in permanent employment, rather, we compare them before and after they enter ITE. Note that the fixed effect coefficients solely reflect changes over time in mental health within the observations for each worker. Compared to the use of prescription drugs for mental health before female workers enter ITE, the model shows an improvement in mental health during the first quarter of temporary employment and for the first quarter before ITE. The decrease in drug use among women with one quarter of ITE is a moderate 1.1 percentage points according to the estimate of -0.011 in column 2 ($p = 0.036$, 95% confidence interval: -0.021 to -0.001). Model 2 in column 2 also shows moderate evidence of a decrease in drug use for women experiencing ITE for four quarters. However, longer periods of ITE appear to be detrimental to mental health as women with six quarters of ITE have significantly higher drug use.

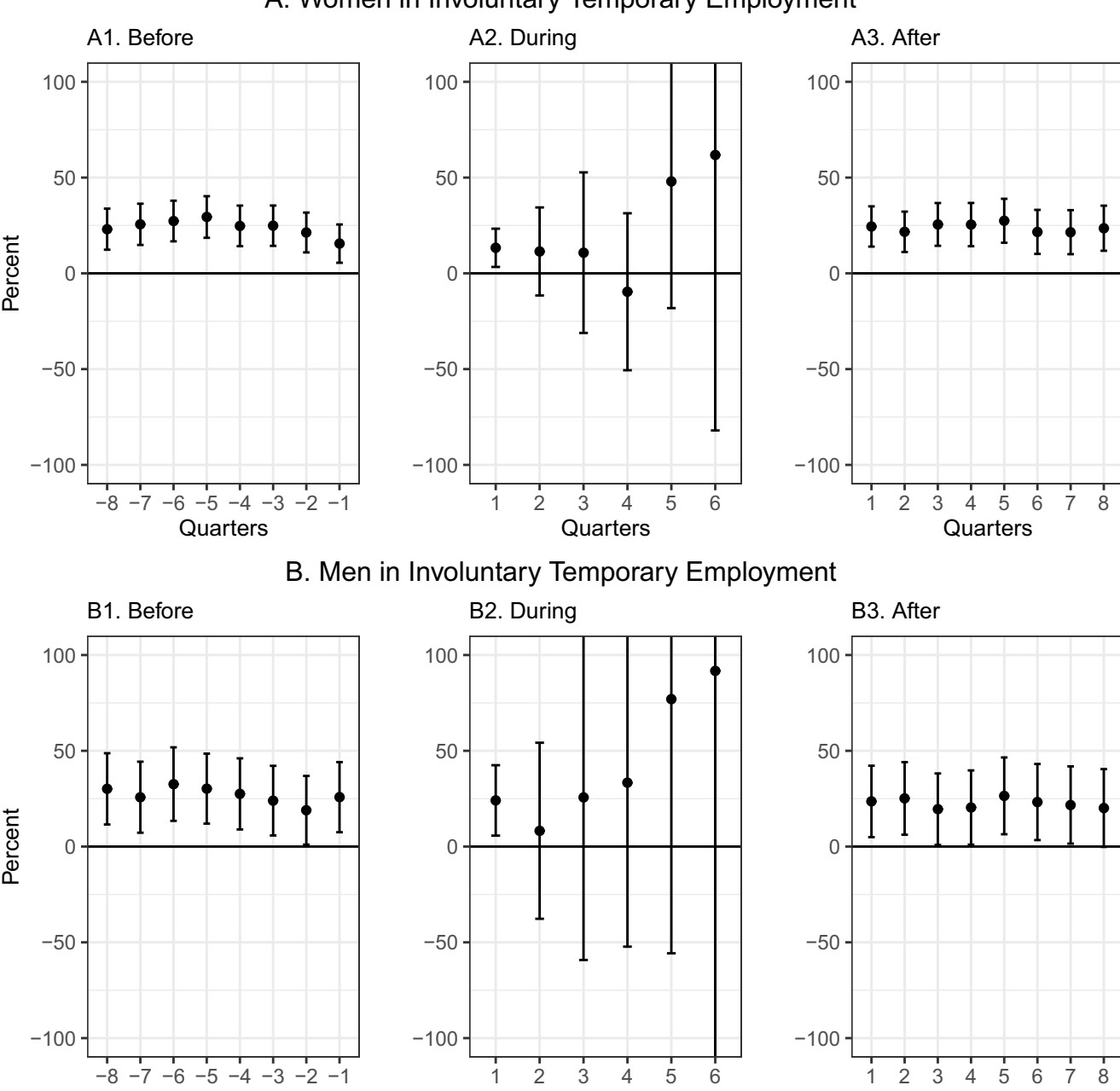

**Fig 2. Quarterly use of prescription drugs before, during, and after involuntarily temporary employment relative to permanent employment for women and men, age adjusted.** Quarters are presented with negative values before involuntary temporary employment to signify the time leading up to the event of employment.

After ITE, the level of drug use returns to the previous level the worker had two to eight quarters before they are observed entering ITE. The exception is women with six quarters of ITE for which a separate indicator provides evidence of increased use of drugs. Women with six observed quarters of ITE appear to experience decreased mental health both during and after ITE. As the effects for women are similar in size during and after six quarters of ITE, we

**Table 2. Involuntary temporary full-time employment and mental health, quarterly observations, 2006–2018.** Dependent variable indicator for drug prescription for mental health problems.

| | WOMEN | | MEN | |
|---|---|---|---|---|
| | **(1)** | **(2)** | **(3)** | **(4)** |
| Involuntary temporary employment | 0.0219*** (0.0052) | - | 0.0108** (0.0052) | - |
| 1 quarter before involuntary temporary employment | -0.0075*** (0.0033) | -0.0083** (0.0039) | -0.0005 (0.0035) | 0.0018 (0.0038) |
| Length of involuntary temporary employment | | | | |
| Quarter 1 | -0.0053** (0.0044) | -0.0109** (0.0052) | -0.0011 (0.0045) | 0.0006 (0.0048) |
| Quarter 1–2 | -0.0019 (0.0127) | -0.0041 (0.0075) | -0.0125 (0.0147) | -0.0119* (0.0072) |
| Quarter 1–4 | -0.0617*** (0.0244) | -0.0319* (0.0168) | 0.0166 (0.0374) | 0.0006 (0.0197) |
| Quarter 1–3 | -0.0038 (0.0281) | 0.0140 (0.0195) | 0.0197 (0.0378) | 0.0188 (0.0138) |
| Quarter 1–6 | 0.0632 (0.0708) | 0.1244** (0.0553) | 0.0337 (0.0727) | -0.0033 (0.0093) |
| After involuntary temporary employment | 0.0032 (0.0039) | -0.0009 (0.0059) | -0.0021 (0.0038) | 0.0047 (0.0053) |
| After involuntary temporary employment, 1–6 quarters | 0.0460 (0.0824) | 0.1314** (0.0624) | -0.0238 (0.0585) | -0.0359 (0.0955) |
| Education level | | | | |
| Low | 0.0225*** (0.0042) | -0.0153 (0.0218) | 0.0193*** (0.0028) | 0.0160 (0.0212) |
| High | -0.0198*** (0.0029) | -0.0278 (0.0119) | -0.0161*** (0.0024) | 0.0120 (0.0197) |
| White collar | -0.0218*** (0.0028) | -0.0016 (0.0071) | -0.0090*** (0.0022) | -0.0005 (0.0075) |
| Married | -0.0133*** (0.0025) | -0.0103 (0.0099) | -0.0039** (0.0022) | 0.0003 (0.0121) |
| Children, up to 6 years old | -0.0111*** (0.0053) | -0.0107 (0.0179) | 0.0127*** (0.0047) | 0.0225** (0.0145) |
| Children, up to 18 years old | -0.0184*** (0.0029) | 0.0066 (0.0143) | -0.0137*** (0.0024) | -0.0298** (0.0142) |
| Income, normalized | -0.0286*** (0.0031) | 0.0029 (0.0093) | -0.0074*** (0.0014) | -0.0075* (0.0041) |
| Estimation method | OLS | FE | OLS | FE |
| Sample | Whole | Treatment | Whole | Treatment |
| Constant | 0.1312*** (0.0058) | 0.1860*** (0.0166) | 0.0817*** (0.0050) | 0.0973*** (0.0169) |
| No. of workers | 75,507 | 5,365 | 84,043 | 3,827 |

***Notes***

Significance levels: * 10%, ** 5%, *** 1%. Standard errors in parentheses are clustered at the person level and calculated using sample weights. Columns (1) and (3) contain Ordinary Least Squares (OLS) estimates on the sample of workers in involuntary full-time temporary employment (treatment group) and workers in full-time permanent employment (comparison group). Columns (2) and (4) contain Fixed Effects (FE) estimates of workers in involuntary full-time temporary employment (treatment group), where the medication level of the workers is compared with their level before they entered involuntary temporary employment. The indicator for temporary employment for, e.g., "Quarter 1–6" takes the value 1 for each of the 6 quarters of temporary employment and 0 otherwise. The income variable is standardized to have mean zero and standard deviation one. The reference person is an unmarried blue-collar worker without children with middle-level education and average income of age 35–39 in the first month of 2012. The data encompasses persons who had full-time employment in at least one of the LFS surveys in 2006–2018 and include mental health indicators of the participants eight quarters before the first survey and eight quarters after the last survey.

estimate a model that assess the overall magnitude by entering a common indicator for both treatment and post-treatment. The common indicator yields a coefficient of 0.128 ($p$ = 0.025, 95% confidence interval: 0.016 to 0.240), which implies an increase of drug use during and after six quarters of ITE by a substantial 12.8 percentage points. As shown in Table 1, the average drug use among woman temporary workers two quarters before entering ITE is 14.2 pct. Thus an increase of 12.8 percentage points nearly doubles this share, while a decrease of 1.1 percentage point (for only one quarter of temporary employment) amounts to less than a ten percent reduction in the share.

The OLS estimates in Model 1 for women (column 1) and men (column 3) do not take unobserved differences between workers into account. For example, workers who enter involuntary full-time temporary employment may have certain personality traits that interact with the both the outcome variable (mental health prescription drugs) and the treatment variable (ITE). By applying fixed effects estimation, Model 2 for women (column 2) and men (column 4) enable us to take time-invariant unobserved differences between workers (such as personality traits) into account. The estimates in Model 1 of the effect of ITE on mental health will thus be downward biased if workers inclined to use prescription drugs tend to avoid entering ITE. Such a bias might be the reason why the coefficients on ITE in Model 2, column 2 are larger than the ones in Mode1, column 1 for lengths of duration beyond 3 months.

For men, we find no statistical significant effects of involuntary temporary employment on prescription drug use for mental health problems. As shown in the results presented in Table 2 for columns 3 and 4, most of the coefficients on the treatment indicators are much smaller than those for women. Like women, men with ITE use more prescription drugs than men with permanent contracts.

## Age differences

Our models for specific age groups reveal that the association between temporary employment and medication for mental health varies across age. As shown in Table 3, for women below age 30 we find no evidence that longer durations of temporary employment have adverse mental health consequences. In contrast, adverse effects of longer periods of temporary employment on prescription drug use for mental health problems appear to be especially pronounced for women aged 30–39 years.

As shown in Table 2, we find no statistical significant associations between ITE and mental health for men. However, according to S1 Table in the Supporting Information, this average result conceals that for the group of men with six quarters of ITE, drug use diminishes after leaving ITE for those over 40, which is not the case for men under 40.

## Involuntary temporary employment and previous labour force states

Workers might follow different mental health paths depending on the type of the previous labour force state. We therefore confine the analysis to those workers where labour market status is observed before they enter ITE. In practice, this means that for these sensitivity analyses we can at most observe respondents over three survey rounds, as one round indicates their employment before they entered ITE. We distinguish between workers entering involuntary full-time temporary employment from being either (1) full-time employed on a permanent contract, (2) part-time employed on a permanent contract, (3) in part-time temporary employment, or (4) unemployed.

For women entering involuntary full-time temporary employment from part-time temporary employment, our OLS regression indicates significant improvements in mental health during full-time ITE. However, our longitudinal fixed effect regression models yield no

**Table 3. Involuntary temporary full-time employment and mental health for women in different age classes, fixed effects estimation, quarterly observations, 2006–2018.** Dependent variable indicator for drug prescription for mental health problems.

| | AGE GROUP | | | |
|---|---|---|---|---|
| | (1) | (2) | (3) | (4) |
| | 20-29 | 30-39 | 40-49 | 50-64 |
| 1 quarter before involuntary temporary employment | -0.0128** (0.0057) | -0.0123 (0.0076) | -0.0019 (0.0098) | -0.0027 (0.0100) |
| Length of involuntary temporary employment | | | | |
| Quarter 1 | -0.0112 (0.0072) | -0.0253** (0.0105) | -0.0015 (0.0132) | 0.0034 (0.0125) |
| Quarter 1–2 | 0.0044 (0.0124) | -0.0158 (0.0143) | 0.0244 (0.0171) | -0.0389** (0.0166) |
| Quarter 1–4 | -0.0042 (0.0071) | -0.0610* (0.0354) | -0.0410 (0.0410) | -0.0130 (0.0245) |
| Quarter 1–5 | 0.0243 (0.0322) | 0.0237 (0.0359) | -0.0071 (0.0438) | -0.0198 (0.0453) |
| Quarter 1–6 | -0.0033 (0.0113) | 0.2369** (0.1159) | 0.0455 (0.0650) | 0.0569 (0.0718) |
| After involuntary temporary employment | 0.0047 (0.0090) | -0.0128 (0.0118) | 0.0111 (0.0150) | -0.0030 (0.0134) |
| After involuntary temporary employment, quarter 1–6 | 0.0985 (0.0791) | 0.2015 (0.1201) | 0.0891 (0.1319) | 0.0567 (0.0919) |
| Education level | | | | |
| Low | -0.0193 (0.0187) | -0.0095 (0.0944) | -0.0682 (0.0612) | -0.0545* (0.0311) |
| High | -0.0186 (0.0137) | -0.0433 (0.0295) | -0.0541 (0.0407) | -0.0408 (0.0585) |
| White collar | -0.0058 (0.0082) | 0.0011 (0.0182) | 0.0047 (0.0157) | -0.0020 (0.0198) |
| Married | -0.0117 (0.0146) | -0.0053 (0.0186) | -0.0072 (0.0266) | -0.0170 (0.0268) |
| Children up to 6 years old | -0.0926* (0.0471) | 0.0223 (0.0174) | 0.0170 (0.0543) | 0.0000 (.) |
| Children up to 18 years old | 0.0417*** (0.0161) | -0.0330 (0.0384) | -0.0081 (0.0248) | 0.0002 (0.0340) |
| Income, normalized | 0.0079 (0.0154) | 0.0201 (0.0164) | 0.0167 (0.0201) | -0.0358 (0.0241) |
| Constant | 0.1139*** (0.0162) | 0.1587*** (0.0351) | 0.2702*** (0.0362) | 0.3803*** (0.0332) |
| No. of workers | 1,635 | 1,462 | 1,077 | 1,191 |

*Notes*: Significance levels: * 10%, ** 5%, *** 1%. Standard errors in parentheses are clustered at the person level and calculated using sample weights. The indicators for involuntary temporary employment for, e.g., "Quarter 1–6" take the value 1 for each of the 6 quarters of temporary employment and 0 otherwise. The income variable is standardized to have mean zero and standard deviation one. The reference person is an unmarried blue-collar worker without children with middle-level education, average income, and aged 35–39 in the first month of 2012. The data encompasses persons who had full-time employment in at least one of the LFS surveys in 2006–2018 and include mental health indicators of the participants eight quarters before the first survey and eight quarters after the last survey.

significant results for transitions from full-time permanent employment, permanent part-time employment, temporary part-time employment and unemployment (for estimates see S2 Table in the Supporting Information). For men, both our OLS and fixed effects models show indications of improvements in mental health for those entering ITE from part-time permanent employment and unemployment (see S3 Table in the Supporting Information). We also find indications of a deterioration in mental health for those entering ITE from full-time permanent contracts (columns 1 and 5).

### Analyses with control groups

The main results in the study stem from fixed effects longitudinal models that assess the use of prescription drugs for mental health during and after ITE relative to workers' health before entering ITE. In these analyses, we do not distinguish whether the workers were unemployed or came from other types of employment before entering involuntary full-time temporary employment. However, mental health workers in ITE might have changed during the estimation period due to observed or unobserved characteristics of workers even if they had not entered temporary employment.

As a supplement to the fixed effects models, we therefore assess the development of the mental health of temporary workers relative to a control group by applying "difference-in-difference matching" [16]. This procedure has the potential to adjust for the bias in the fixed effects estimates that arises if the use of prescription drugs among temporary workers changes over time for other (unobserved) reasons than changes in type of employment contract. We use workers who enter ITE at a later point in time as control groups (see [14, 15] for more technical details of this procedure). The results from matching models that include control groups are shown in S4 Table in the Supporting Information. Results from these matching models deviate only marginally from the results of the regression models without control groups presented in Table 2.

### Discussion

In this study, we show that ITE of longer duration (at least six quarters) has adverse mental health consequences for women, but not for men. Shorter periods of ITE appear to have minor positive mental health consequences for women, but not for men. We obtain these results by analysing survey data from the European Union LFS (2006–2018), supplemented by high-quality administrative data on use of prescriptions of drugs to alleviate mental health problems provided by Statistics Denmark. Three characteristics of our study makes it different from all or most previous studies in the field: (1) The analysis is confined to workers who are involuntarily employed on a temporary contract, (2) the analysis traces the consequences of different spell lengths of temporary employment, and (3) women and men are analysed separately.

First, to the best of our knowledge, our study is the first to analyse mental health consequences for involuntary temporary workers, that is, temporary contract workers who could not find a permanent contract job. However, we are not the first to emphasize the importance of separate analyses of involuntary temporary employment. For example, Bardasi and Francesconi [10] use survey data to find that workers with seasonal or casual work, or work done under contract for a fixed period of time in the UK did not have long-lasting detrimental mental health effects for either female or male workers. They note that due to data limitations they cannot distinguish between involuntary and voluntary temporary employment and that health effects may even be positive. Drawing on the combination of survey data and Danish administrative registers, our study documents that short-term ITE can in fact have positive health effects for the workers' mental health but that long-term temporary employment can have negative consequence for female workers.

Second, our study is–to our knowledge–also the first to analyse mental health consequences of different periods of ITE. Using the method of inverse-probability-of-treatment weights, Pirani and Salvini [31] find that longer periods of temporary employment in Italy have negative consequences for the "general health" of women, but not for men. Our study adds to this line of research by showing that long-term effects also apply for ITE and mental health.

Third, we report gender specific results, which is not common in the literature. In a survey of atypical employment (including temporary employment) and health (including mental health), Gray et al. [13] found only 12 studies published since 2009 that reported gender specific results.

Among the limitations of the study is that employment status is self-reported and not observed outside the survey quarters (information on involuntary employment is, however, difficult to obtain without asking workers). This design has at least two implications for the interpretation of our results. First, we show that women with six quarters of involuntary temporary employment experience adverse mental health. This finding is based on survey data containing two quarters of temporary employment, then two quarters without survey data, and then two quarters of temporary employment. As workers might have been in non-temporary employment in the six months without surveys, the treatment indicator may contain measurement errors that attenuate the coefficients of the treatment in the regressions. Therefore, we view our measures of the adverse mental health consequence of longer periods of temporary employment as conservative estimates.

Second, our data do not contain information about employment before the first survey and it is therefore likely that some workers have been in temporary employment before the first observed quarter. If short-term ITE improves mental health for some workers (as our study shows), this improvement may take place before the first observation of ITE. If so, the coefficient on the indicator for the period before the first observation of temporary employment tends to be negative, which is indeed the case (Table 2, Models 1–2). Another potential reason for this negative coefficient is self-selection, which is the case when some women (due to different life circumstances, such as the rules for receiving welfare benefits) enter ITE when they experience improved mental health.

## Conclusion

Our analysis contains five main results. First, compared to workers on permanent employment contracts, workers on temporary contracts had worse mental health before entering ITE. This result holds true for both women and men. Second, entering temporary contracts did not appear to have adverse effects on mental health in the short run. On the contrary, for women we found a minor improvement in mental health at the beginning of this type of employment contract. Third, for women we found substantial adverse effects of ITE on mental health for temporary contracts for six quarters. Fourth, after leaving ITE, we found no permanent effects of shorter spells of temporary employment on use of prescription drugs for mental health problems. Women with longer periods of temporary employment experienced worse mental health after the termination of ITE. Fifth, for women we found evidence that the improvement in mental health during involuntary temporary full-time work stems from women coming from other types of atypical employment.

However, more studies are needed to draw definitive conclusions about whether the adverse effects are smaller in the Scandinavian countries compared to other countries. In addition, as we find different patterns for men compared to women with regard to sign and magnitude of the effects, future research should investigate gender differences in the effects of involuntary temporary employment.

## Supporting information

**S1 Table. Involuntary temporary full-time employment and mental health for men in different age classes, fixed effects estimation, quarterly observations, 2006–2018.**
(DOCX)

**S2 Table. Women in involuntary temporary full-time employment and mental health according to labour market state before temporary employment, quarterly observations, 2006–2018.**
(DOCX)

**S3 Table. Men in involuntary temporary full-time employment and mental health according to labour market state before temporary employment, quarterly observations, 2006–2018.**
(DOCX)

**S4 Table. Involuntary temporary full-time employment and mental health compared to control group, quarterly observations, 2006–2018.**
(DOCX)

## Author Contributions

**Conceptualization:** Karsten Albæk, Stefan Bastholm Andrade.

**Data curation:** Karsten Albæk, Stefan Bastholm Andrade.

**Formal analysis:** Karsten Albæk, Stefan Bastholm Andrade.

**Funding acquisition:** Karsten Albæk, Stefan Bastholm Andrade.

**Investigation:** Karsten Albæk, Stefan Bastholm Andrade.

**Methodology:** Karsten Albæk, Stefan Bastholm Andrade.

**Project administration:** Karsten Albæk, Stefan Bastholm Andrade.

**Resources:** Karsten Albæk, Stefan Bastholm Andrade.

**Software:** Karsten Albæk, Stefan Bastholm Andrade.

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
