## [Decision Letter · Decision Letter 0]

18 Jun 2023

PGPH-D-23-00532

Involuntary temporary work and mental health in Denmark

Dear Dr. albæk,

Thank you for submitting your manuscript to PLOS Global Public Health. After careful consideration, we feel that it has merit but does not fully meet PLOS Global Public Health’s publication criteria as it currently stands. Therefore, we invite you to submit a revised version of the manuscript that addresses the points raised during the review process.

Please see the comments from two reviewers below and in the attachment. Both reviewers find the study and its contribution interesting, but have provided a number of comments to help strengthen and clarify the message. Reviewer 1's comments about consistent and clear definitions throughout are particularly pertinent, and will aid readers in understanding how the findings can be used.

We look forward to receiving your revised manuscript.

Kind regards,

Hanna Landenmark

Staff Editor

Journal Requirements:

1. Our staff editors have determined that your manuscript is likely within the scope of our Global Mental Health: challenges, opportunities, and the future of the field. This editorial initiative is headed by a team of Guest Editors for PLOS GPH: Rochelle Burgess (University College of London) and Dixon Chibanda (University of Zimbabwe and London School of Tropical Medicine and Hygiene). The Collection invites researchers to submit original research which engages with, or disrupts, the urgent needs across the global mental health landscape. We especially encourage submissions of studies that critically interrogate the status quo of the field and that involve inter-/trans-disciplinary approaches and those which share perspectives from underrepresented global regions and communities.

 Additional information can be found on our announcement page: https://collections.plos.org/call-for-papers/global-mental-health-opportunities-challenges/ 

If you would like your manuscript to be considered for this collection, please let us know in your cover letter and we will ensure that your paper is treated as if you were responding to this call.  Please note that being considered for the Collection does not require additional peer review beyond the journal’s standard process and will not delay the publication of your manuscript if it is accepted by PLOS GPH. If you would prefer to remove your manuscript from collection consideration, please specify this in the cover letter.

Additional Editor Comments (if provided):

Reviewers' comments:

Reviewer's Responses to Questions

**Comments to the Author**

1. Does this manuscript meet PLOS Global Public Health’s publication criteria? Is the manuscript technically sound, and do the data support the conclusions? The manuscript must describe methodologically and ethically rigorous research with conclusions that are appropriately drawn based on the data presented.

Reviewer #1: Partly

Reviewer #2: Yes

2. Has the statistical analysis been performed appropriately and rigorously?

Reviewer #1: Yes

Reviewer #2: Yes

3. Have the authors made all data underlying the findings in their manuscript fully available (please refer to the Data Availability Statement at the start of the manuscript PDF file)?

Reviewer #1: No

Reviewer #2: No

4. Is the manuscript presented in an intelligible fashion and written in standard English?

Reviewer #1: Yes

Reviewer #2: Yes

5. Review Comments to the Author

Reviewer #1: Review: PGPH-D-23-00532

Very interesting and innovative research topic.

Some points need to be tackled to improve the manuscript:

Title:

Adjust the title to make it more specific about your research topic and the research methods.

My suggestion:

Involuntary temporary work and mental health medications: a longitudinal study in Denmark

Abstract;

Make somewhere already clear that you only selected full-time workers.

Throughout manuscript (and abstract):

Pay attention to your wording; this is an article about “Involuntary” full-time temporary employment, yet throughout the description of your results and in the discussion, the word “involuntary” is often omitted making the manuscript confusing and hard to follow (as the reader is believed to think that the results are about temporary employment (independent of being involuntary). Add the word “involuntary” to the text where necessary.

The authors highlight too much that their study is on mental health. Their study is only on the use of medication for certain mental health problem. Many people who have (common) mental health problems do not use medication. The authors should be more transparent about how specific their study is and use the wording “the use of medication for mental health problems” instead of mental health as much as possible.

Introduction:

P 2: 3 paragraph: “this is the first study to do a separate analysis” – what do you mean by “separate analysis”. Lots of studies are done about the health effects of temporary employment.

More information (e.g. previous research) should also be given about what is already know about medication for mental health problems and temporary employment.

Methodology

Add more information about the data, what are the years that these data were gathered?

More information is also needed about who is selected into the group that are analyzed: there age-range.

A visualization of the data and used gathered would be a useful tool to understand the data better. The authors report on a “pause” in the collecting of the survey. This will also become more clear how this affected the data and results, with a visualization.

Be more specific about what a quarter means (is this 3 month?) – could also be helpful in the tables.

Can you control your results for income and family composition, as these are an important confounder between employment and mental health. The authors should think about more variable which could be added as controls.

Describe also what methodology is used for the descriptive tables.

Describe whether you used weights, how you handled missing data, whether some variables had high missing data.

In the end you do an analysis with part-time employed; this should already be explained in the methodology.

Results

This will be an article which will be interesting to many occupational sociologists. The use of the medical words “Treatment group” and “pre-treatment”, “post treatment”, etc. is difficult to follow when not being from a medical background. These words should be replaced by the “labels” which they stand for “such as, permanent, and involuntary temporary employment. This will make the interpretation of the results, tables and figure much more clarity.

Tables

Avoid to use of the words “quarter” it is very difficult to understand.

Figure & Tables:

What is pre-treatment? Unemployed?

Post-treatment? Not being involuntary temporary employed anymore?

Discussion

Be specific when you mention “shorter periods” – how many months are that?

P 13, first sentence above: is that compared to fulltime permanent workers? Please be specific.

The authors mention that “since the mid-1990 “non-standard” jobs increased. However, within the occupational sociology, scholars usually mention the 1970’s as the start of more flexible contracts.

See Benach et al 2014 Precarious Employment: Understanding an Emerging Social Determinant of Health. Annual Review of Public Health

P14 Last sentence of the discussion: “When some women choose to enter temporary employment” is this really an issue in your study when you only selected “involuntary” temporary workers?

Conclusion:

Second sentence: “workers on temporary contract has worse mental health before entering “temporary employment ….

This sentence is confussing - what do you mean? What was there status then before they entered temporary employment? Unemployed?

Reviewer #2: The main aim of the paper is to bridge the gap in the literature with regard to the relationship between involuntary temporary employment and mental health. The authors estimate gendered effects using various estimation measures and assess the medium-term effect of being in involuntary employment. In general, the paper is well-written and conveys its main messages clearly. In the attached document, I have added my detailed comments and suggestions.

6. PLOS authors have the option to publish the peer review history of their article (what does this mean?). If published, this will include your full peer review and any attached files.

**Do you want your identity to be public for this peer review?** For information about this choice, including consent withdrawal, please see our Privacy Policy.

Reviewer #1: No

Reviewer #2: No

---

## [Editor Report · Decision Letter 1]

6 Sep 2023

PGPH-D-23-00532R1

Involuntary temporary work and mental health medications: A longitudinal study in Denmark

Dear Dr. albæk,

Thank you for submitting your manuscript to PLOS Global Public Health. After careful consideration, we feel that it has merit but does not fully meet PLOS Global Public Health’s publication criteria as it currently stands. Therefore, we invite you to submit a revised version of the manuscript that addresses the points raised during the review process.

Dear authors,

First of all, I was reviewer 1 of the previous version of your article and PLOS Global Public Health now asked me to be the guest editor for your manuscript.

I carefully read your revised manuscript and the letters to the reviewers and believe your article has improved a lot. From the beginning I believed this is a very interesting and well-performed researchpaper for occupational researchers.

Yet, I still see some points which need clarification and mentioning in the discussion. My comments are also based on the comments of Reviewer 2.

Before, accepting your manuscript to PLOS Global Public Health, I would like you to address the following:

1) Reviewer's 2 comment number 13 (Only in the discussion that we read about the fact that the temporary employed could have started before the first quarter, when the individual is first captured in the data. In light of this the authors should be careful with the use of the terms short and long-term results of temporary employment and entering temporary employment. This issues deserves thorough discussion in the methods section).

I'm not satisfied with the adjustments made by the authors (only using months to describe the duration).

I would like to see, just like Reviewer 2, a discussion about this issues in the methods section already and how this would affect your results.

Moreover, in the sensitivity analysis The authors mention, we confine the analysis to those workers where the labour market status is observed before entering "involuntary temporary employment" (ITE). Why is it possible that the authors can observe the labour market status before entering ITE, and at the same time the temporary employed could have started before the first quarter? Can you not also exclude those who started ITE before the first quarter then?

2) Reviewer's 3 comment number 3 (The authors excluded part-time employees and only included in their sample full-time employees. A reasoning is needed for excluding this category especially since women usually represent a large percentage of part-time workers).

I believe the authors are not answering reviewer's 2 comment correctly. Reviewer 2 is talking about part-time work and the authors are answering about temporary or short-term employment, this is not the same. Could you please mention in the methods section already what the reasoning was to exclude part-time workers and how this would affect your results (since women usually are part-time workers).

3) Figure 1 needs some more work. It is still very difficult to understand the way the data is gathered. In the manuscript it is talked about 4 waves and in the figure I see 6 rounds? Could you also add the years (2006, 2018, etc...) to your figure, name your x and y-axis, as a reader I want to be able to understand the "months" which the article mentions so much.

4) Sensitivity analyses (Involuntary temporary employment and previous labour force status). Please do not call part-time employment or voluntary temporary employment "precarious employment". This is not at all in line with consensus about precarious employment. Part-time employment can be "protected" employment because of good employment conditions, the same holds for temporary employment. Precarious employment can be considered a multidimensional construct and only measuring it with 1 is not the best way to do it. Group 2 talking part-time workers and voluntary temporary workers together is not the best idea? split them up?

5) Please read your article carefully for typo's.

6) Table 2, Model 2 and 4, for the sake of clarity please write in the notes the comparison group of these models.

Best regards,

Deborah De Moortel

We look forward to receiving your revised manuscript.

Kind regards,

Deborah De Moortel

Guest Editor
---

## [Editor Report · Decision Letter 2]

27 Oct 2023

Involuntary temporary work and mental health medications: A longitudinal study in Denmark

PGPH-D-23-00532R2

Dear Senior researcher albæk,

We are pleased to inform you that your manuscript 'Involuntary temporary work and mental health medications: A longitudinal study in Denmark' has been provisionally accepted for publication in PLOS Global Public Health.

Best regards,

Deborah De Moortel

Guest Editor

Thank you for this well-done revision! This paper is very interesting and statistical analyses are well done and well explained!